# Morphology and Chemical Coding of Rat Duodenal Enteric Neurons following Prenatal Exposure to Fumonisins

**DOI:** 10.3390/ani12091055

**Published:** 2022-04-19

**Authors:** Katarzyna Kras, Halyna Rudyk, Siemowit Muszyński, Ewa Tomaszewska, Piotr Dobrowolski, Volodymyr Kushnir, Viktor Muzyka, Oksana Brezvyn, Marcin B. Arciszewski, Ihor Kotsyumbas

**Affiliations:** 1Department of Animal Anatomy and Histology, University of Life Sciences in Lublin, 12 Akademicka St., 20-950 Lublin, Poland; katarzyna.kras@up.lublin.pl; 2State Scientific Research Control Institute of Veterinary Medicinal Products and Feed Additives, Donetska St. 11, 79000 Lviv, Ukraine; galusik.77@gmail.com (H.R.); wolodjak@gmail.com (V.K.); muzyka@scivp.lviv.ua (V.M.); brezvun@gmail.com (O.B.); dir@scivp.lviv.ua (I.K.); 3Department of Biophysics, Faculty of Environmental Biology, University of Life Sciences in Lublin, 13 Akademicka St., 20-950 Lublin, Poland; siemowit.muszynski@up.lublin.pl; 4Department of Animal Physiology, Faculty of Veterinary Medicine, University of Life Sciences in Lublin, 12 Akademicka St., 20-950 Lublin, Poland; ewaRST@interia.pl; 5Department of Functional Anatomy and Cytobiology, Faculty of Biology and Biotechnology, Maria Curie-Sklodowska University, Akademicka St. 19, 20-033 Lublin, Poland; piotr.dobrowolski@mail.umcs.pl

**Keywords:** mycotoxins, enteric nervous system, neuropeptides, small intestine, rat

## Abstract

**Simple Summary:**

Food contamination with toxins produced by various species of fungi (hereafter: mycotoxins) occurs frequently, and, for this reason, their activity in the bodies of animals is the subject of wide-ranging research. Although many aspects of the activity of fumonisins (mycotoxins produced by *Fusarium*) have been described, an open question is whether prenatal exposure to fumonisins may result in morphological changes, including in the enteric nervous system (ENS). The present study revealed that fumonisin administered via the gastrointestinal tract to pregnant rats did not substantially change the structure of the intestine/ENS of the offspring, but altered the neurochemical profile of its enteric neurons.

**Abstract:**

Fumonisins (FBs), including fumonisin B1 and B2 produced by the fungus *Fusarium verticillioides*, are widespread mycotoxins contaminating crop plants as well as processed food. The aim of the experiment was to determine whether the exposure of 5-week-old pregnant rats to FBs at 60 mg/kg b.w. (group FB_60_) or 90 mg/kg b.w. (group FB_90_) results in morphological changes in the duodenum of weaned offspring, particularly the enteric nervous system (ENS). In addition, the levels of expression of galanin and vasoactive intestinal polypeptide (VIP) in the ENS were analysed by immunofluorescence in the control and experimental groups of animals. No significant morphological changes in the thickness of the muscle layer or submucosa of the duodenum were noted in group FB_60_ or FB_90_. In group FB_90_ (but not FB_60_), there was a significant increase in the width of the villi and in the density of the intestinal crypts. Immunofluorescence analysis using neuronal marker Hu C/D showed no significant changes in group FB_60_ or FB_90_ in the morphology of the duodenal ENS, i.e., the myenteric plexus (MP) and submucosal plexus (SP), in terms of the density of enteric ganglia in the MP and SP, surface area of MP and SP ganglia, length and width of MP and SP ganglia, surface area of myenteric and submucosal neurons, diameter of myenteric and submucosal neurons, density of myenteric and submucosal neurons, and number of myenteric and submucosal neurons per ganglion. In both groups, there was an increase (relative to the control) in the percentage of Hu C/D-IR/VIP-IR (IR-immunoreactive) and Hu C/D-IR/galanin-IR myenteric and submucosal neurons in the ganglia of both the MP and SP of the duodenum. In addition, in groups FB_60_ and FB_90_, there was an increase in the number of nerve fibres showing expression of VIP and galanin in the mucosa, submucosa and circular muscle layer of the duodenum. The results indicate that prenatal exposure to FBs does not significantly alter the histological structure of the duodenum (including the ENS) in the weaned offspring. The changes observed in the chemical code of the myenteric and submucosal neurons in both experimental groups suggest harmful activity of FBs, which may translate into activation of repair mechanisms via overexpression of neuroprotective neuropeptides (VIP and galanin).

## 1. Introduction

In the 1970s, increased incidence of equine leukoencephalomalacia was observed in South Africa, which was eventually linked to the consumption of maize contaminated with fungi. The main fungus species isolated from contaminated maize was *Fusarium verticillioides*, which thereafter attracted the attention of researchers [1,2]. In subsequent years, a correlation was observed between the consumption of contaminated maize and increased occurrence of oesophageal cancer in humans [3], which prompted researchers to search for and isolate new substances with mycotoxin activity. In 1988, Gelderblom et al. extracted, purified and chemically tested two substances isolated from *Fusarium verticillioides* strain MRC 826 with carcinogenic potential, which were called fumonisin B1 (FB1) and fumonisin B2 (FB2) [4]. Subsequently, mass spectrometry and high-resolution nuclear magnetic resonance were used to describe the exact structure of the isolated FBs [5]; for review, see also [6]. FBs have been shown to be similar in their structure and activity to sphingolipids, and both substances act as ceramide synthase inhibitors and cause accumulation of bioactive intermediates of sphingolipid metabolism [7,8]. Given that sphingolipids are ubiquitous cell membrane components involved in a variety of cellular processes, such as signal transduction and cell survival [9,10], the potentially harmful effects of FBs in mammals are of particular importance. Moreover, it has recently been found that intoxication with FBs may result in induction of oxidative stress [11,12], autophagy activation [13], or even induction of epigenetic changes [14,15]. The numerous mechanisms of action of FBs may explain why, despite their very poor absorption following oral administration and thus limited accumulation in tissues [16], these mycotoxins cause substantial changes in many mammalian systems, particularly the digestive organs [8,17]. It should be noted that, due to its function, it is the gastrointestinal tract (especially the initial parts of the small intestine) that is most exposed to direct contact with mycotoxins. A number of functional changes have been observed in the small intestine of pigs and rats experimentally intoxicated with FBs, including villi flattening, atrophy and fusion, changes in the size of the enteric ganglia, segment-dependent reduction of goblet cell numbers, lymphatic infiltrates, and disturbance of the permeability of the intestinal barrier [18,19,20,21,22]. It has also been postulated that FB intoxication changes the proportions of gut microbiota [23]. While that study primarily focused on the effects of FB administration on postnatal development and in adults, it must be emphasized that FBs are classified as a potentially teratogenic factor [24]. In newborns which were potentially exposed to FB as foetuses via administration of the mycotoxin to the mother, various changes in the structure of the gastrointestinal tract were observed [25], as well as neural tube defects (NTDs) [26]. However, it is not entirely clear to what extent FB administered orally crosses the placental barrier of pregnant females and to what degree it is neurotoxic to developing foetuses.

Therefore, the aim of the present study was to use immunohistochemical staining to investigate the potentially toxic effects of FBs on the structure of the small intestine (duodenum) and the corresponding enteric nervous system (ENS) of weaned rats whose mothers were treated orally with various doses of mycotoxin during pregnancy. Additionally, taking into account the presence of mechanisms of neuroplasticity in the ENS, we investigated whether the administration of FBs influenced expression levels of neuroprotective neuropeptides (galanin and vasoactive intestinal polypeptide, VIP) in the enteric neurons of the duodenum of weaned rats.

## 2. Materials and Methods

### 2.1. Fumonisins 

FB1 and FB2 were produced as previously described [27] from a culture of *Fusarium verticillioides* in a maize grain medium. Briefly, autoclaved coarsely cracked maize seeds were inoculated with *F. verticillioides* cultured in Petri dishes with tryptone glucose yeast extract broth (from the biobank at the Laboratory of Mycotoxicology, Institute of Veterinary Medicine of the National Academy of Agrarian Sciences of Ukraine, Kiev, Ukraine) and kept in a dark place (24 °C) for the next 4 weeks. Next, the incubation was stopped and the maize seeds were collected and autoclaved at 121 °C for 15 min. Then, the grains were dried at 80–90 °C for 120 min, ground, and stored at −20 °C. Liquid chromatography analysis showed that the FB1:FB2 ratio was 3:1 (73% to 27%). FBs were extracted from dried seeds with ethyl alcohol and analysed by ELISA. Finally, after determining the concentration, the extract was evaporated to obtain a concentration of 100 mg/mL.

### 2.2. Animals 

The study was conducted in compliance with the guidelines of the Declaration of Helsinki and approved by the Institutional Ethics Committee of the State Scientific Research Control Institute of Veterinary Medicinal Products and Feed Additives in Lviv, Ukraine (#132 676-Adm/08/2020, 28 February 2020). The experiment was conducted on pregnant 5-week-old Wistar rats (n = 18) receiving a mixture of FB1 and FB2 from the 7th day of pregnancy (to omit the period of organogenesis).

Rats were housed individually in polypropylene cages (380 × 200 × 590 mm) and underwent a one-week acclimatization period to become accustomed to the laboratory conditions. They were kept at a temperature of 21 ± 3 °C, humidity of 55 ± 5%, with a 12 h/12 h day/night cycle, and had free access to water.

After the acclimatization period, the pregnant rats were randomly divided into three groups: one control (n = 6) and two experimental (FB_60_ and FB_90_, n = 6 each). All animals were fed ad libitum with mycotoxin free standard diet for laboratory rodents. Pregnant rats from group FB_60_ were daily intoxicated by intragastric administration of FB1 + FB2 (3:1 ratio) at a dose of 60 mg/kg b.w., whereas animals from group FB_90_ (n = 6) received 90 mg/kg b.w. of FB1 + FB2 (3:1 ratio) in the same way. The FB mixture was administered in 0.5 mL of 0.9% saline solution until parturition. The administered dose of 90 mg FB1 + FB2/kg b.w. was equal to 1/10 of the established LD_50_ value, which was sufficient to induce subclinical intoxication [22], while the dose of 60 mg FB1 + FB2/kg b.w. was equal to 1/15 of the LD_50_ value. Rats from the control group (n = 6) received saline solution in a corresponding amount and manner. After birth, the offspring were culled to the same number of offspring in each litter and kept with their mothers until weaning at 28 days. One randomly selected weaned rat (28 days old) from each litter in every group fasted for 24 h and was euthanized by CO_2_ inhalation.

### 2.3. Tissue Processing

In each control and experimental rat, the abdomen was opened with a midline incision and the duodenum was located. Starting 1.0 cm from the gastric pylorus, a duodenal fragment 2.0 cm in length was collected from each animal. This material was carefully washed in saline solution (37 °C) and transferred to a cold buffered formalin solution (4 °C, pH = 7.4) for 24 h. Then, the fixed duodenum was cut into two 1 cm parts (one for morphometric analysis and one for immunofluorescence analysis).

The material for immunofluorescent (IF) staining was washed twice in 0.01 M phosphate buffered saline (PBS) for 10 min and then transferred to a transparent container filled with 16% sucrose solution (4 °C) supplemented with 0.01% bacteriostatic sodium azide (Avantor Performance Materials Poland S.A., Gliwice, Poland). The sucrose solution was replaced with fresh solution every day until the material fell to the bottom of the container. Subsequently, immediately after the final washing in PBS, the material was embedded in Tissue-Tek^®^ O.C.T.™ Compound (Sakura Finetek USA, Inc., Torrance, CA, USA) and frozen in dry ice. Then, the frozen sections were cut to a thickness of 8 µm with a cryostat (HM 525 NX, Thermo Scientific, Waltham, MA, USA). Every fourth section was placed on adhesive glass slides (Superfrost Plus, Thermo Scientific, Waltham, MA, USA), and finally the slides were stored at −20 °C until further IF analysis. The remaining duodenal fragments, intended for morphometric analyses, were first dehydrated and cleared in Ottix Plus and Ottix Sharper (DiaPath, Martinengo, Italy) and then saturated with paraffin using a tissue processor (STP 120, Thermo Scientific, Waltham, MA, USA). Then, the processed material was embedded in paraffin blocks using the EC 350 automatic sample preparation system (Especialidades Médicas Myr S.L., Tarragona, Spain). Finally, the paraffin blocks were cut into 4 μm thick sections with a microtome (Microm HM 360, Microm, Walldorf, Germany). Every fourth section was placed on adhesive glass slides (Superfrost Plus, Thermo Scientific) and kept in an incubator (CG Wamed, Warsaw, Poland) at 37 °C for 12 h.

### 2.4. Immunofluorescence

Selected slides were first warmed at room temperature (RT) for 30 min, and then the sections were outlined with a hydrophobic marker (ImmEdge™ Hydrophobic Barrier Pen, Vector Laboratories, Burlingame, CA, USA). Next, the sections were washed in 0.01 M PBS with 0.25% Triton X-100 (Sigma-Aldrich, Saint Louis, MO, USA) (3 × 10 min), covered with bovine serum albumin to block non-specific protein binding sites, and washed again. Then, solutions of primary antibodies (details of antisera are listed in Table 1) were dropped onto the slides and left for overnight incubation in a dark humid chamber (RT). Mouse anti Hu C/D antibodies (used as a pan-neuronal marker for visualization of enteric neurons) were combined with either rabbit antisera raised against VIP or against galanin. The next day, the excess antibody solution was removed, and the slides were again washed in PBS and incubated for 1 h with a combination of species-specific and fluorochrome-conjugated secondary antibodies. Following incubation, the slides were washed for the last time, mounted in phosphate-buffered glycerol (pH = 8.2), and finally coverslipped. Two different procedures were used to test the specificity of the antibodies. In the first step, considered a negative control, sections not exposed to primary antibodies (omitted or replaced with non-immunoreactive sera) were stained. The second procedure was a pre-absorption experiment in which primary antibodies were mixed with an excess of target synthetic protein before incubation. No positive immunoreaction was detected in any of the control sections.

### 2.5. Cell Counting, Imaging and Statistical Analysis

Immunofluorescence-labelled sections were viewed with an epifluorescence microscope (BX61 Olympus, Nagano, Japan) equipped with interference filter cubes optimized for detection of Alexa Fluor 595 (MWIY2, excitation/emission wavelength 545–580 nm; Olympus, Tokyo, Japan) and Alexa Fluor 488/FITC (MNIBA2, excitation/emission wavelength 470–490 nm; Olympus). Images were captured using Cell^M 2.3 software (Olympus cellSens Standard) and a digital camera (C11440-36U, Hamamatsu Photonics, Shizuoka, Japan) connected to a standard PC. Images were taken under a 20× objective at a resolution of 1024 × 1024 pixels. Each stained section was examined in its entirety, and expression of biologically active substances was assessed by analysing all the ganglia seen on the cross-section of the duodenum of each animal (but no fewer than 100 myenteric and submucosal neurons per animal). Further morphometric analysis of the duodenal ENS structure was performed using ImageJ 1.52 software [28]. The following parameters were measured for myenteric and submucosal ganglia: neuron area, neuron diameter, ganglion area, ganglion length, ganglion width, number of ganglia per mm, number of neurons per mm, and number of neurons per ganglion. Next, the percentages of galanin-immunoreactive (galanin-IR) and VIP-immunoreactive (VIP-IR) perikarya were calculated and expressed as a percentage of the total numbers of Hu C/D-IR myenteric/submucosal neurons. The densities of galanin-IR and VIP-IR nerve fibres were also counted using ImageJ and a grid mask (each square of 1000 μm^2^). The grid mask was superimposed on the original images, and the number of nerve fibres per square was counted. The densities of galanin-IR and VIP-IR fibres were also estimated on a semi-quantitative scale: very numerous (>2.51 IR nerve fibres per 1000 μm^2^), numerous (2.1–2.5), moderate (1.51–2), few (0.01–1.5) and absent (0) [29].

### 2.6. Morphometry

Slides for morphometric analysis were dewaxed with xylene and rehydrated with descending grades of ethyl alcohol. Goldner’s trichrome staining was applied to differentiate the layers of duodenal wall [30]. The analyses were performed using ImageJ software [28]. The thickness of the muscular layer, mucosa and submucosa was measured using a straight line tool (30 measurements per parameter on each duodenum slice). The length and width of undamaged intestinal villi and the length and width of the crypts were measured using a segmented line tool (10 measurements per parameter on each duodenum slice). Additionally, the numbers of villi and crypts per mm were calculated. An experimental unit was a single rat. Measurements were averaged per animal prior statistical analysis

### 2.7. Statistical Analysis

The data were analysed using Statistica 13.3 software (TIBCO Software Inc., 2017; Palo Alto, CA, USA). Parameters were compared between the control group and the experimental groups. One-way analysis of variance (ANOVA) was performed, followed by Tukey’s honestly significant difference (HSD) test. The normality of data distribution was tested using the Shapiro–Wilk test, and homogeneity of variance was tested with Levene’s test. Statistically significant differences were assumed for *p* < 0.05. Results are expressed as means ± standard deviation.

## 3. Results

### 3.1. ENS Morphology

Immunostaining with Hu C/D antibodies enabled precise identification of enteric neurons (the presence of Hu C/D was confined to the neuronal soma, neuroplasm and nuclei), as neuronal perikarya stained using this neuronal marker were bright and clearly distinguishable against the dark background (see Figure 1 and Figure 2). Administration of FBs at 60 and 90 mg/kg b.w. during pregnancy did not significantly change either the mean numbers of myenteric ganglia per mm or the mean numbers of myenteric neurons per ganglion (see Table 2). No statistical changes (vs. control) in the geometry (mean area, length, and width) of the single myenteric ganglion were observed in either group FB_60_ or group FB_90_. The mean area and diameter of a single Hu C/D-IR myenteric neuron did not change after treatment during pregnancy with either 60 or 90 mg/kg b.w. Neither the mean numbers of submucosal ganglia nor the numbers of Hu C/D-IR submucosal neurons per ganglion changed significantly (vs. control) in group FB_60_ or FB_90_ (see Table 3). The administration of FBs during pregnancy at 60 mg/b.w. did not affect the area, length or width of the submucosal ganglia. Similarly, FBs given during pregnancy at 90 mg/b.w. did not cause any significant differences (vs. control) in the dimensions of submucosal ganglia. In animals from group FB_60_ and group FB_90_, there were no significant differences in the mean area or diameter of Hu C/D-IR submucosal neurons in comparison to the controls.

### 3.2. Morphometric Measurements of the Intestine

The use of Goldner’s trichrome staining method enabled visualization of individual histological layers in the duodenal wall of the control and experimental animals (Figure 3). In animals from FB_60_ and FB_90_ groups, the mean thickness of the duodenal longitudinal smooth muscle layer and the circular muscle layer were not significantly changed when compared to controls (see Table 4). The mean thickness of the submucosal layer in animals from group FB_60_ and FB_90_ was not significantly different to that of the control animals. Similarly, administration of different doses of FBs during pregnancy had no effect on the mean thickness of the mucosal layer. In group FB_60_, the mean density of the mucosal villi and the mean villus height and width were not changed vs. controls. However, although in group FB_90_ the mean numbers of villi per mm of intestine remained unchanged to controls, the mean villus width was significantly greater. No statistical differences in the mean height of the villi of animals from control and FB_90_ group were found. Animals from group FB_90_ had a significantly higher density of duodenal crypts than the controls, but this effect was not observed in group FB_60_. There were no statistically significant differences between the crypt dimensions (mean depth and width) in the control, FB_60_ and FB_90_ groups.

### 3.3. Immunofluorescence 

In the control animals, immunofluorescent staining revealed a small subpopulation of Hu C/D-IR myenteric duodenal neurons expressing VIP (see Table 5). FBs administered during pregnancy at 60 and 90 mg/kg b.w. resulted in a significant increase in VIP in Hu C/D-IR myenteric neurons. The proportions of Hu C/D-IR/VIP-IR submucosal neurons were higher in both experimental groups and these values were statistically significant when compared to controls. VIP was also detected in nerve fibres supplying the intestinal layers of animals in the control and experimental groups. Administration of FBs caused a statistically significant increase in the numbers of VIP-IR nerve fibres in the circular muscle layer, submucosal layer and mucous layer in both groups (see Figure 1).

In the control animals, the presence of galanin was noted in both myenteric and submucosal neurons. Prenatal administration of FBs at both doses significantly changed the subpopulation of Hu C/D-IR/galanin-IR myenteric neurons in both groups. Similarly, an increase in numbers of Hu C/D-IR/galanin-IR submucosal neurons was observed in groups FB_60_ and FB_90_. The administration of FBs at 60 and 90 mg/kg b.w. during pregnancy significantly changed the mean density of galanin-IR nerve fibres supplying the circular muscle layer, the submucosa and the mucosa (see Figure 2).

## 4. Discussion

The results of the study clearly showed that FBs administered to pregnant rats (in two different doses) did not cause significant morphological or measurable changes in the layer structure of the duodenum of their offspring, but they did induce certain changes in the duodenal ENS. Previously published research reports suggest factors that may potentially influence the effect of FBs in the prenatal period. First, it is clear that previous studies have used various doses of FBs. Due to the ubiquity of FBs in food for people (cereals including rye, wheat, barley and rice) and animal feed (maize and maize-based products), many research centres have attempted to establish safe levels of FB intake that would not negatively affect animal health, but the results are often conflicting. As insufficiently decontaminated feed may contain many similarly acting mycotoxins at the same time, their synergistic and at times additive effects must be taken into account, which makes determination of maximum levels even more difficult [31]. According to FDA recommendations from 2001, safe levels of FBs (FB1 + FB2 + FB3) for animals depend on the species and purpose of the animal, as livestock animals, like horses, sheep or pigs, or laboratory animals like rodents are more sensitive to FB when compared to, for example, poultry, and range from 5 to 100 mg/kg of feed, with a median of 25 and an average of 37.5 mg/kg feed [32]. The levels of FBs used in our experiment (60 and 90 mg/kg b.w.) amount to 1/15 and 1/10 of the LD_50_ value for FBs, determined in a preliminary study as 900 mg/kg b.w. for rats. Importantly, both levels were higher than those inducing NTDs in the offspring of pregnant mice receiving FBs intraperitoneally from 7.5 and 8.5 days of gestation [33].

The existence of various mechanisms of action of FBs on developing foetuses has been postulated. One of these assumes inhibition of sphingolipid synthesis, which leads to the accumulation of intermediate metabolites, and this in turn can disturb signalling cascades involved in embryonic development, including proliferation, differentiation and migration of cells [34,35]. Therefore, it can also be presumed that, after FBs pass through the placenta of the pregnant female, the most likely effect should be disturbance of embryonic or foetal development. Inhibition of sphingolipid synthesis may also indirectly lead to disturbances in the supply of folic acid to the developing foetus [24], as well as to depletion of glycosphingolipid reserves, which in turn can disturb the expression and function of folate receptor 1, FOLR1 [36]. Folic acid is commonly known to play a key role in nervous system development, and thus blockage of its transfer to the foetus generally results in NTDs. This relationship between administration of FB and the development of NTDs has been confirmed in many studies, e.g., using rat embryos exposed to the toxin at levels from 0.2 to 217 ppm [37] and mouse embryos in a range from 0.71 to 71.2 ppm [38], but also in studies in which FB was administered to pregnant mice at levels from 2.5 to 100 mg/kg b.w. [8,33,39]. Increased incidence of NTDs has also been observed in human foetuses whose mothers’ diet consisted mainly of maize products contaminated with *F. verticillioides* [36].

Another factor considered by many researchers to be the main cause of disturbances in the development of foetuses potentially exposed to FBs in utero is maternal toxicity. Under that assumption, penetration of FBs through the placental barrier is not obvious. Several studies using pregnant rats, hamsters, mice and rabbits did not show NTDs, but other developmental disturbances were observed, involving a reduction in body weight [38,40,41,42], deformations [43] or increased foetal mortality [42,43], which is largely explained by the secondary effects of FBs via maternal toxicity. These studies also showed that sensitivity to various amounts of FB is species-dependent. In most cases, as in the present study, FB was administered beginning at 7 to 9 days of gestation, omitting the organogenesis period, but—what is important in light of the present study—the start of administration coincided with the migration of nerve stem cells colonizing the digestive tract and ultimately forming the ENS. This is because ENS development in rodents begins relatively late; cells derived from the neural crest do not begin to migrate towards the gastrointestinal tract until day 9.5 of embryonic development [44]. Normal ENS development depends on numerous factors and is regulated at several levels. The large number of processes regulating ENS development and thus the large number of signalling molecules enable potential interactions between these molecules and exogenous compounds to which foetuses may be exposed if the compounds penetrate the placental barrier [45]. In addition, nervous system cells largely owe their special structure and function to sphingolipids, which help to preserve the normal physiology of the neuron or oligodendrocyte, including their differentiation, polarization, and in the case of neurons, synapse formation and synaptic conduction [46].

To our surprise, however, in the present study, we did not observe significant changes in ENS structure in either the MP or the SP. Previous experiments have shown that the addition of FBs to the diet can significantly affect the morphology of ENS neurons. Specifically, administration of feed with the addition of FBs to rats for 42 days reduced the surface area of neurons in both the SP and the MP of the jejunum [47]. In another experiment, 21-day intragastric intoxication of 5-week-old rats with FBs significantly reduced the surface area of ENS ganglia in the duodenum (in this case only in the SP) and the jejunum (in the SP and MP) [22]. Other mycotoxins are known to have similar effects on ENS morphology as in the case of FBs. For example, deoxynivalenol (DON) administered orally to adult rats for 42 days reduced the area of muscle neurons and the area of MP ganglia in the jejunum [48]. While there are few reports of ENS damage caused by mycotoxins, there are numerous reports concerning the effects of these toxins on the structure of gastrointestinal tract organs and their consequences. A number of mycotoxins produced by *Fusarium* spp. (besides FBs, these include DON, T2 toxin and zearalenone (ZEN)), as well as others such as patulin, ochratoxin or aflatoxin, are usually absorbed directly through the mucosa of individual parts of the digestive tract and induce classical symptoms of food poisoning such as diarrhoea and vomiting [49,50]. Direct contact of the mucosa of the small intestine with the toxin may explain the changes observed in its histological structure, i.e., decreased villus length, disruption of the epithelial barrier, changes in the number of goblet cells, and the presence of local inflammation combined with intensified apoptosis [49,51]. In our study, however, the small intestinal wall exhibited none of the structural changes typically associated with mycotoxins. This may be explained in part by the route of administration of the toxin, as the animals were exposed to FBs via umbilical cord blood, bypassing the gastrointestinal tract. The changes we observed in the histological structure of the duodenum included an increase in the width of the villi and the density of crypts in group FB_90_, which suggests that the harmful effect of FBs administered in the prenatal period was partially dose-dependent.

Although no morphological changes were observed in the ENS, there seems to have been a reaction to the harmful activity of FBs via neuroplasticity. Harmful factors (e.g., inflammation, axotomy or toxins) are known to cause neurons to produce and release certain neuroprotective substances, such as VIP and galanin [52,53,54,55]. It should be borne in mind that these neuropeptides also take part physiologically in the regulation of the motility and secretory activity of the intestines [56,57]. In addition, in pathological states, apart from the neuroprotective function mentioned above, VIP is involved in immunomodulation of the intestines, while galanin modulates inflammatory processes within the intestines [49,52,53]. Moreover, it has been suggested that increased expression of galanin may result from its role in tissue regeneration following inflammation or organ damage [52]. These observations are consistent with the results of the present study, as FB intoxication caused an increase in the percentages of both VIP-positive and galanin-positive neurons in both the MP and the SP of the duodenum, as well as an increase in the density of VIP-positive and galanin-positive nerve fibres in the circular muscle layer, submucosa, and mucosa of the duodenum. Similar results have been obtained in studies determining the effects of other mycotoxins on chemical coding of ENS neurons. Oral administration of T2 toxin and ZEN to immature pigs for 42 days resulted in an increase in the percentage of VIP-positive neurons in both the SP and MP and an increase in the density of VIP-positive nerve fibres in the circular muscle layer and mucosa of the small intestine [49]. Interestingly, immature pigs receiving ZEN orally for 42 days showed a significant decrease in the density of galanin-positive nerve fibres in the circular muscle layer of the ileum [58], which suggests that this toxin may primarily have impaired the peristalsis of the intestine. These last results are in contrast with those obtained in our study, but it should be noted that the route of administration of toxins was different in the two experiments (via the placenta vs. via the digestive tract).

## 5. Conclusions

The results of the study indicated that prenatal exposure to FBs via administration to pregnant rats in the amount of 60 mg/kg b.w. and 90 mg/kg b.w. does not significantly affect the histological structure of the duodenum of weaned rats, although certain changes, i.e., a decrease in villus width and crypt density, were seen in the case of the higher dose. Prenatal exposure to FBs (at both doses) also had no significant effect on the morphology of the duodenal ENS; however, changes were noted in the chemical code of ENS neurons, which can be linked to the neuroprotective activity of galanin and VIP in response to the harmful activity of FB.

## Figures and Tables

**Figure 1 animals-12-01055-f001:**
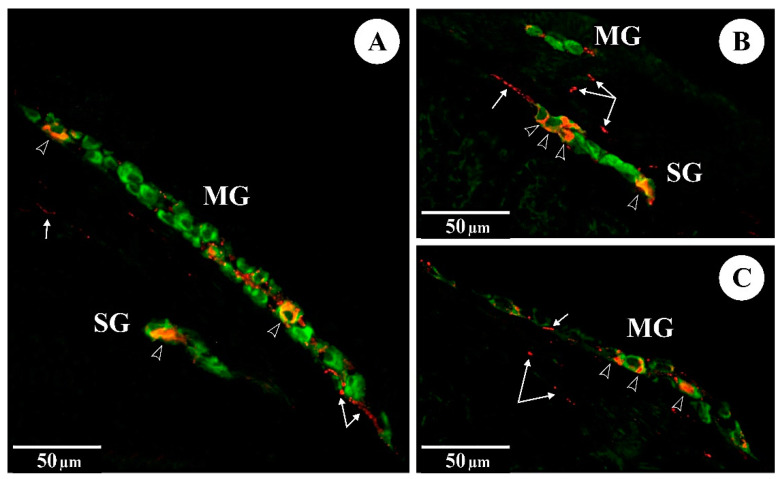
The immunoreactivity of Hu C/D (green) and VIP (red) in myenteric (MG) and submucosal (SG) ganglia of the rat duodenum: (**A**) control group; (**B**) FB_60_ group; (**C**) FB_90_ group; empty arrowheads indicate Hu C/D-IR/VIP-IR neurons, arrows indicate VIP-IR fibres.

**Figure 2 animals-12-01055-f002:**
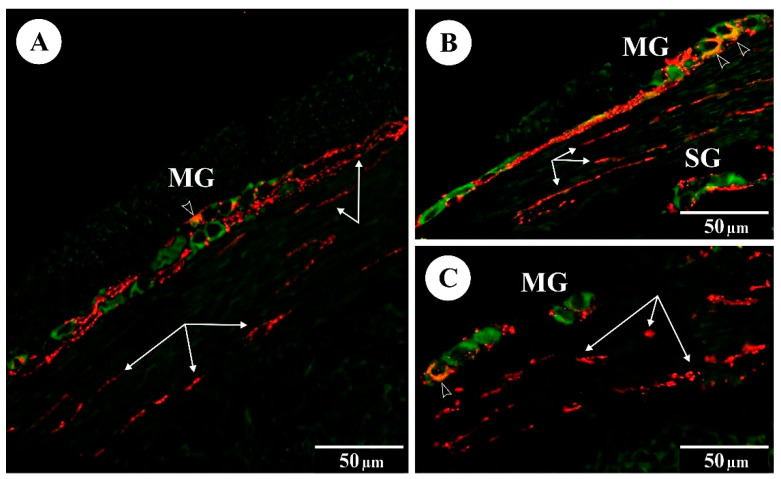
The immunoreactivity of Hu C/D (green) and galanin (red) in myenteric (MG) and submucosal (SG) ganglia of the rat duodenum: (**A**) control group; (**B**) FB_60_ group; (**C**) FB_90_ group; empty arrowheads indicate Hu C/D-IR/galanin-IR neurons, arrows indicate galanin-IR fibres.

**Figure 3 animals-12-01055-f003:**
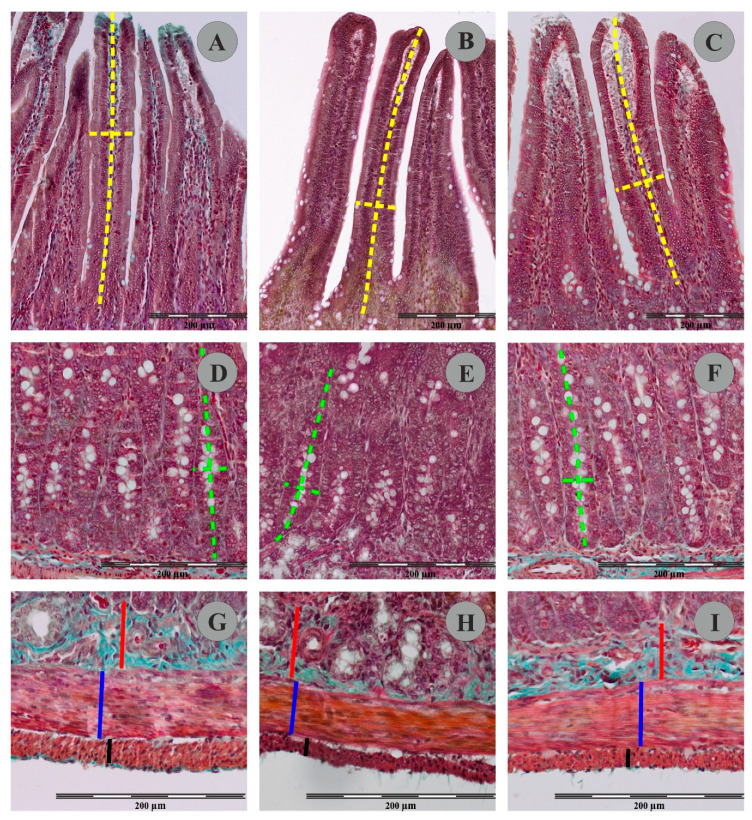
Representative pictures of villus (**A**–**C**); crypts (**D**–**F**) muscular and submucosal layer (**G**–**I**) in Goldner’s trichrome stained sections of rat duodenum from the control group (**A**,**D**,**F**), FB_60_ group (**B**,**E**,**H**) and FB_90_ group (**C**,**F**,**I**). Please note measurement schemes of the length (a long dashed yellow line) and the thickness (a short dashed yellow line) of the villus, the depth (a long green dashed line) and the width (a short green dashed line) of the crypts, the thickness of the submucosal layer (a continuous red line) and the thickness of the the circular muscle layer (a continuous blue line) and the thickness of the longitudinal muscle layer (a continuous black line).

**Table 1 animals-12-01055-t001:** Characteristics of primary and secondary antisera used for immunofluorescence experiments.

Primary Antibody	Host	Dilution	Code	Source
Anti-Human Neuronal Protein Hu C/Hu D (anti-Hu C/D)	Mouse	1:400	A-21271	Molecular Probes, Eugene, OR, USA
Anti-galanin	Rabbit	1:1400	4600–5004	Biogenesis, Paterson, NJ, USA
Anti-vasoactive intestinal peptide (VIP)	Rabbit	1:1200	ab22736	Abcam, Cambridge, UK
Anti-mouse Alexa Fluor 488	Donkey	1:800	A11029	Thermo Fisher Scientific, Waltham, MA, USA
Anti-rabbit Alexa Fluor 594	Donkey	1:800	A21207	Thermo Fisher Scientific, Waltham, MA, USA

**Table 2 animals-12-01055-t002:** Effect of the fumonisins intoxication on the morphology of the rat duodenal myenteric plexus (mean ± SD).

Parameter	Group			*p*-Value
	Control	FB_60_	FB_90_	
Neuron area (µm^2^)	122.83 ± 15.98	114.88 ± 12.93	96.17 ± 11.18	n.s.
Neuron diameter (µm)	11.18 ± 1.16	11.21 ± 1.03	10.56 ± 0.98	n.s.
Ganglion area (µm^2^)	864.21 ± 297.79	786.33 ± 169.95	726.76 ± 281.17	n.s.
Ganglion length (µm)	92.70 ± 32.15	73.22 ± 17.80	76.68 ± 24.36	n.s.
Ganglion width (µm)	13.18 ± 2.11	12.71 ± 1.26	12.03 ± 2.32	n.s.
Ganglia (per 1 mm)	1.84 ± 0.84	1.59 ± 0.50	1.37 ± 0.64	n.s.
Neurons per ganglion	5.10 ± 2.66	5.31 ± 2.46	4.97 ± 1.09	n.s.

Statistical significance: n.s.—not significant.

**Table 3 animals-12-01055-t003:** Effect of the fumonisins intoxication on the morphology of the rat duodenal submucosal plexus (mean ± SD).

Parameter	Group			*p*-Value
	Control	FB_60_	FB_90_	
Neuron area (µm^2^)	123.15 ± 44.50	117.86 ± 20.87	110.19 ± 15.53	n.s.
Neuron diameter (µm)	11.56 ± 1.56	11.21 ± 1.03	11.41 ± 1.07	n.s.
Ganglion area (µm^2^)	412.58 ± 78.73	370.70 ± 84.75	336.72 ± 57.80	n.s.
Ganglion length (µm)	44.60 ± 12.86	39.72 ± 9.49	32.97 ± 7.17	n.s.
Ganglion width (µm)	13.07 ± 3.47	11.02 ± 2.71	12.28 ± 1.81	n.s.
Ganglia (per 1 mm)	1.41 ± 0.64	1.02 ± 0.68	0.71 ± 0.17	n.s.
Neurons per ganglion	2.10 ± 0.62	1.89 ± 0.27	1.76 ± 0.45	n.s.

Statistical significance: n.s.—not significant.

**Table 4 animals-12-01055-t004:** Effect of the fumonisin intoxication on the rat duodenum morphology (mean ± SD).

Parameter	Group			*p*-Value
	Control	FB_60_	FB_90_	
Longitudinal muscle layer thickness (µm)	23.49 ± 4.41	21.06 ± 4.71	19.43 ± 4.46	n.s.
Circular muscle layer thickness (µm)	52.18 ± 8.22	42.56 ± 6.38	40.77 ± 8.34	n.s.
Submucosa thickness (µm)	27.04 ± 5.77	30.42 ± 11.88	27.59 ± 6.05	n.s.
Mucosa thickness (µm)	819.32 ± 94.53	753.71 ± 41.71	825.51 ± 53.65	n.s.
Villus height (µm)	589.70 ± 71.70	557.55 ± 57.70	623.26 ± 37.37	n.s.
Villus width (µm)	75.90 ± 6.28 ^a^	76.59 ± 3.56 ^a^	100.85 ± 15.76 ^b^	<0.001
Density of villi (mm^−1^)	8.24 ± 1.41	6.39 ± 1.12	8.82 ± 1.38	n.s.
Crypt depth (µm)	229.63 ± 38.71	196.16 ± 30.07	202.25 ± 31.13	n.s.
Crypt width (µm)	36.23 ± 3.89	39.18 ± 3.28	31.82 ± 3.40	n.s.
Density of crypts (mm^−1^)	22.99 ± 2.80 ^a^	21.16 ± 2.35 ^a^	26.79 ± 1.54 ^b^	0.002

Statistical significance: n.s.—not significant; Different letters indicate significant differences between groups.

**Table 5 animals-12-01055-t005:** Effects of fumonisins intoxication on the chemical coding of the rat duodenal enteric neurons (mean ± SD).

Parameter	Group			*p*-Value
	Control	FB_60_	FB_90_	
VIP-IR myenteric neurons	11.61 ± 1.61 ^a^	15.14 ± 2.35 ^b^	17.65 ± 4.10 ^b^	0.008
VIP-IR submucosal neurons	24.84 ± 3.35 ^a^	27.46 ± 2.55 ^b^	30.23 ± 3.08 ^b^	0.024
VIP-IR fibres				
Muscular layer	0.48 ± 0.07 ^a^	0.58 ± 0.07 ^b^	0.66 ± 0.06 ^b^	<0.001
Submucosa	0.23 ± 0.04 ^a^	0.30 ± 0.06 ^b^	0.32 ± 0.06 ^b^	0.041
Mucosa	0.43 ± 0.04 ^a^	0.77 ± 0.07 ^b^	0.81 ± 0.12 ^b^	<0.001
galanin-IR myenteric neurons	21.05 ± 2.84 ^a^	27.47 ± 4.85 ^b^	30.02 ± 5.90 ^b^	0.014
galanin-IR submucosal neurons	19.21 ± 3.35 ^a^	25.97 ± 4.76 ^b^	27.06 ± 4.74 ^b^	0.014
galanin-IR fibres				
Muscular layer	0.83 ± 0.12 ^a^	0.94 ± 0.12 ^b^	1.11 ± 0.07 ^b^	<0.001
Submucosa	0.13 ± 0.04 ^a^	0.23 ± 0.07 ^b^	0.25 ± 0.05 ^b^	0.004
Mucosa	0.12 ± 0.04 ^a^	0.17 ± 0.05 ^b^	0.19 ± 0.01 ^b^	0.017

Statistical significance: n.s.—not significant; Different letters indicate significant differences between groups.

## Data Availability

Not applicable.

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
