# Peer review of "Morphology and Chemical Coding of Rat Duodenal Enteric Neurons following Prenatal Exposure to Fumonisins"

_animals, 2022, doi:10.3390/ani12091055_

Round 1

Reviewer 1 Report

The manuscript brings results about prenatal exposure of rats to Fusarium-produced mycotoxins (fumonisins). The authors showed that fumonisin
given via the gastrointestinal tract to pregnant rats did not change the intestine structure of the offspring, however altered the neurochemical profile of its enteric neurons. These findings indicate neuroprotective activity of galanin and vasoactive intestinal polypeptide in response to the harmful activity of fumonisins.

The manuscript is well structured, well designed and well discussed. I have no further suggestion to make. Congratulations on the hard and interesting work.

Author Response

Thank you very much for your review and agreeing that manuscript is ready for publication.

Reviewer 2 Report

The work is experimental in nature and is carried out using modern research methods. The number of animals used in the study is appropriate. There are some shortcomings in the work that require correction:

In some parts of the manuscript galanin is shortened to “gal” (line 191, 193, 195) whereas in others not. This should be unified.

Throughout the bodytext the name of a neuronal marker used is written as “Hu C/D”, “HuCD” or “HuC/D”. Please correct this.

Line 42 – It needs to be explained what IR stands for.

Line 55 – Fusarium verticillioides should be written in italic.

Line 122 – also ad libitum should be written in italic.

Line 144 – please provide a producer of Tissue-Tek® O.C.T.™ Compound.

Line 147 – please change “(Superfrost Plus, Thermo Scientific)” to “(Superfrost Plus, Thermo Scientific, Waltham, MA, USA)” and remove “Waltham, MA, USA” from the line 151

Line 161 – “table 1” should be started with a capital letter

Line 236 – as there are no significant differences the value 0.56±0.98 µm appears to be incorrect

Line 264 – “Masson Goldner’s” or  “Masson-Goldner’s” ???

Line 355 – NTDs is abbreviated for the second time. First time was at line 342 but first time when it was introduced is in line 86.

Line 359 – [38, 8, 32] should be changed to [8, 32, 38].

Please check and unify the format of listed references. For example: please remove “;” after the last author name in references 16, 18

Reference 29 – please check the title. It should rather be “Connective and mesenchymal tissues with their stains” but not “12 - Connective and other mesenchymal tissues with their stains”

Author Response

In some parts of the manuscript galanin is shortened to “gal” (line 191, 193, 195) whereas in others not. This should be unified. Corrected

Throughout the bodytext the name of a neuronal marker used is written as “Hu C/D”, “HuCD” or “HuC/D”. Please correct this. Corrected as suggested.

Line 42 – It needs to be explained what IR stands for. Appropriate explanation has been added.

Line 55 – Fusarium verticillioides should be written in italic. Corrected

Line 122 – also ad libitum should be written in italic. Corrected

Line 144 – please provide a producer of Tissue-Tek® O.C.T.™ Compound. Sakura manufacturer has been indicated in the text.

Line 147 – please change “(Superfrost Plus, Thermo Scientific)” to “(Superfrost Plus, Thermo Scientific, Waltham, MA, USA)” and remove “Waltham, MA, USA” from the line 151. Corrected

Line 161 – “table 1” should be started with a capital letter. Corrected

Line 236 – as there are no significant differences the value 0.56±0.98 µm appears to be incorrect. Thank you for pointing out this obvious mistake. We have corrected this value to 10.56±0.98

Line 264 – “Masson Goldner’s” or “Masson-Goldner’s” ??? Thank you for your valuable comment. After a second thought, we think that “Goldner’s trichrome” will be the most appropriate term.

Line 355 – NTDs is abbreviated for the second time. First time was at line 342 but first time when it was introduced is in line 86. Corrected

Line 359 – [38, 8, 32] should be changed to [8, 32, 38]. Corrected

Please check and unify the format of listed references. For example: please remove “;” after the last author name in references 16, 18. We have once again carefully checked the references and  minor mistakes noticed have been corrected. During the revision we noticed that some important reports concerning neuroprotective role of VIP and galanin were also missing. This issue has also been fixed.

Reference 29 – please check the title. It should rather be “Connective and mesenchymal tissues with their stains” but not “12 - Connective and other mesenchymal tissues with their stains”. Corrected

Reviewer 3 Report

This is a well written and overall well-argued manuscript describing some very interesting results of histological and immunohistological analysis of duodenum of rats prenatally exposed to fumonisins (FBs), which indicate harmful activity of FBs to developing fetus. However, you should address some aspects, which in my opinion require your attention.

  1. Please make sure throughout the manuscript that all Fusarium species are written italic (for example L21, L55-56, L61).
  2. While the discussion finishes with an interesting paragraph describing the effects of mycotoxins on intestines and ENS in livestock animals, I suggest adding in the introduction a very short information about the prevalence of feed/food infections with fumonisins/mycotoxins.
  3. Materials and methods: For all equipment, please provide the complete information about manufacturer and model type if necessary.
  4. Please do not write that some parameters were “statistically comparable“ or “remained statistically similar” between groups, but that there was no statistical significance between the (mean) values observed in groups.
  5. Please consider presenting all the data in tables. It would make easier for the readers to observe the noted statistically significant changes. Please also give the exact p-values for data where the statistically significant changes were observed.

L64 “[5],for review see also [6].

L102 [27] shows the different sources of the F. vericillioides. Please correct if necessary.

L107 could you provide the exact values?

L122 was the diet checked for the presence of fumonisins and other mycotoxins?

L125 daily?

L130 Please specify whether the litters were culled to the same number of offspring in each or not.

L132 the results of rat's body weight are not presented. Please remove the information about weighting or provide appropriate data.

L182 pixels

L186, 203 Please unify the data about ImageJ. Also, it is recommended to use a proper reference, please see ImageJ homepage for details.

L209 What was the experimental unit? Single rat? Were measurements averaged per animal prior statistical analysis?

L232 “width 12.71±1.26 um”

L290 “Goldner stained sections of rat duodenum”

L353 Correct to "folate receptor 1, FOLR1".

Author Response

1/ Please make sure throughout the manuscript that all Fusarium species are written italic (for example L21, L55-56, L61).

The format of latin names has been once again checked and corrected.

2/ While the discussion finishes with an interesting paragraph describing the effects of mycotoxins on intestines and ENS in livestock animals, I suggest adding in the introduction a very short information about the prevalence of feed/food infections with fumonisins/mycotoxins.

Thank you very much for this important remark. We agree that brief clarification of food contamination with mycotoxin (especially fumonisins) will be beneficial for the present study. As we believe the Introduction is a fairly consistent chapter as it stands we would like to add this information in discussion (Lines 362-363)

3/ Materials and methods: For all equipment, please provide the complete information about manufacturer and model type if necessary.

We have provided requested information concerning chemicals and equipment used.

4/ Please do not write that some parameters were “statistically comparable“ or “remained statistically similar” between groups, but that there was no statistical significance between the (mean) values observed in groups.

Thank you very much for this valuable comment. This issue was clarified and necessary corrections have been done.

5/ Please consider presenting all the data in tables. It would make easier for the readers to observe the noted statistically significant changes. Please also give the exact p-values for data where the statistically significant changes were observed.

We took into account your remark. Now numerical data are presented in tables which also include the exact p-values (when statistically significant).

L64 “[5],for review see also [6]. Corrected

L102 [27] shows the different sources of the F. vericillioides. Please correct if necessary. Corrected

L107 could you provide the exact values? Exact values have been added as suggested

L122 was the diet checked for the presence of fumonisins and other mycotoxins? Yes, the necessary information has been added in the text.

L125 daily? Yes, it was clarified in the text.

L130 Please specify whether the litters were culled to the same number of offspring in each or not.  It is indeed important information. The litters were culled to the same number of offspring in each. The necessary information has been added.

L132 the results of rat's body weight are not presented. Please remove the information about weighting or provide appropriate data. Removed as suggested.

L182 pixels. Added

L186, 203 Please unify the data about ImageJ. Also, it is recommended to use a proper reference, please see ImageJ homepage for details. Corrected

L209 What was the experimental unit? Single rat? Were measurements averaged per animal prior statistical analysis? Single rat was the experimental unit and measurements were averaged per animal prior statistical analysis. The requested information have been added to the manuscript.

L232 “width 12.71±1.26 um”. Corrected

L290 “Goldner stained sections of rat duodenum”. Corrected

L353 Correct to "folate receptor 1, FOLR1". Corrected